# Following the niche: the differential impact of the last glacial maximum on four European ungulates

Michela Leonardi [1✉], Francesco Boschin [2✉], Paolo Boscato[2] & Andrea Manica [1]

Predicting the effects of future global changes on species requires a better understanding of the ecological niche dynamics in response to climate; the large climatic fluctuations of the last 50,000 years can be used as a natural experiment to that aim. Here we test whether the realized niche of horse, aurochs, red deer, and wild boar changed between 47,000 and 7500 years ago using paleoecological modelling over an extensive archaeological database. We show that they all changed their niche, with species-specific responses to climate fluctuations. We also suggest that they survived the climatic turnovers thanks to their flexibility and by expanding their niche in response to the extinction of competitors and predators. Irrespective of the mechanism behind such processes, the fact that species with long generation times can change their niche over thousands of years cautions against assuming it to stay constant both when reconstructing the past and predicting the future.

[1] Evolutionary Ecology Group, Department of Zoology, University of Cambridge, Downing Street, Cambridge CB2 3EJ, UK. [2] U.R. Preistoria e Antropologia, Dipartimento di Scienze Fisiche della Terra e dell'Ambiente, Università degli Studi di Siena, Via Laterina 8, 53100 Siena, Italy. ✉email: ml897@cam.ac.uk; francesco.boschin@unisi.it

Given the current rate of climatic changes due to human activity[1], there is an urgent need to establish the best possible framework to study how animal species react to climate change and how their ecological niche may vary over time. The climate fluctuations that characterized the last tens of thousands of years can be used as a virtual lab to understand better the ecological niche dynamics under different environmental conditions, which, in turn, may help define better conservation strategies for the future[2].

The last 50 thousand years (ky) provide an ideal period since it is characterized by considerable climatic fluctuations[3]. Looking at Europe, this period encompasses both the Last Glacial Maximum, when ice covered almost half of the continent and the Holocene climatic amelioration that led to a drastic change in overall vegetation composition[4]. The last 50 ky also present a significant technological advantage: they are covered by the radiocarbon dating method ([14]C [5],). This allows gathering observations from the archaeological record with precise chronological attribution and associating them with palaeoclimatic reconstructions. In addition, the recent development of nearly-continuous palaeoclimatic data series over the last tens of thousands of years[6–8]

provides the appropriate context to investigate species responses to climatic change over this time scale.

Species Distribution Models (SDMs)[9] allow characterizing the realized niche of the species[10] by associating its occurrences with environmental variables of the area they inhabit (Fig. 1), producing a model that can predict its potential geographic distribution based on suitable climate.

When SDMs have been applied to palaeoecological databases to reconstruct the niche through time, they have often been fitted independently to each time slice with enough occurrences and for which paleoclimate reconstructions were available (e.g[11–15]). The estimations for each time slice can then be compared to each other to detect possible changes[12] (Fig. 1B). However, this approach can be problematic since the number of occurrences available for any time slice is often limited. This problem is further compounded because the limited sampling has not equal effort across the range, giving a geographically incomplete coverage (see the discrepancies between the sampled points and the outlined distribution in Fig. 1A). Figure 1B illustrates how applying SDMs independently to each time slice may lead to substantial errors in the result, as a geographic bias in the samples

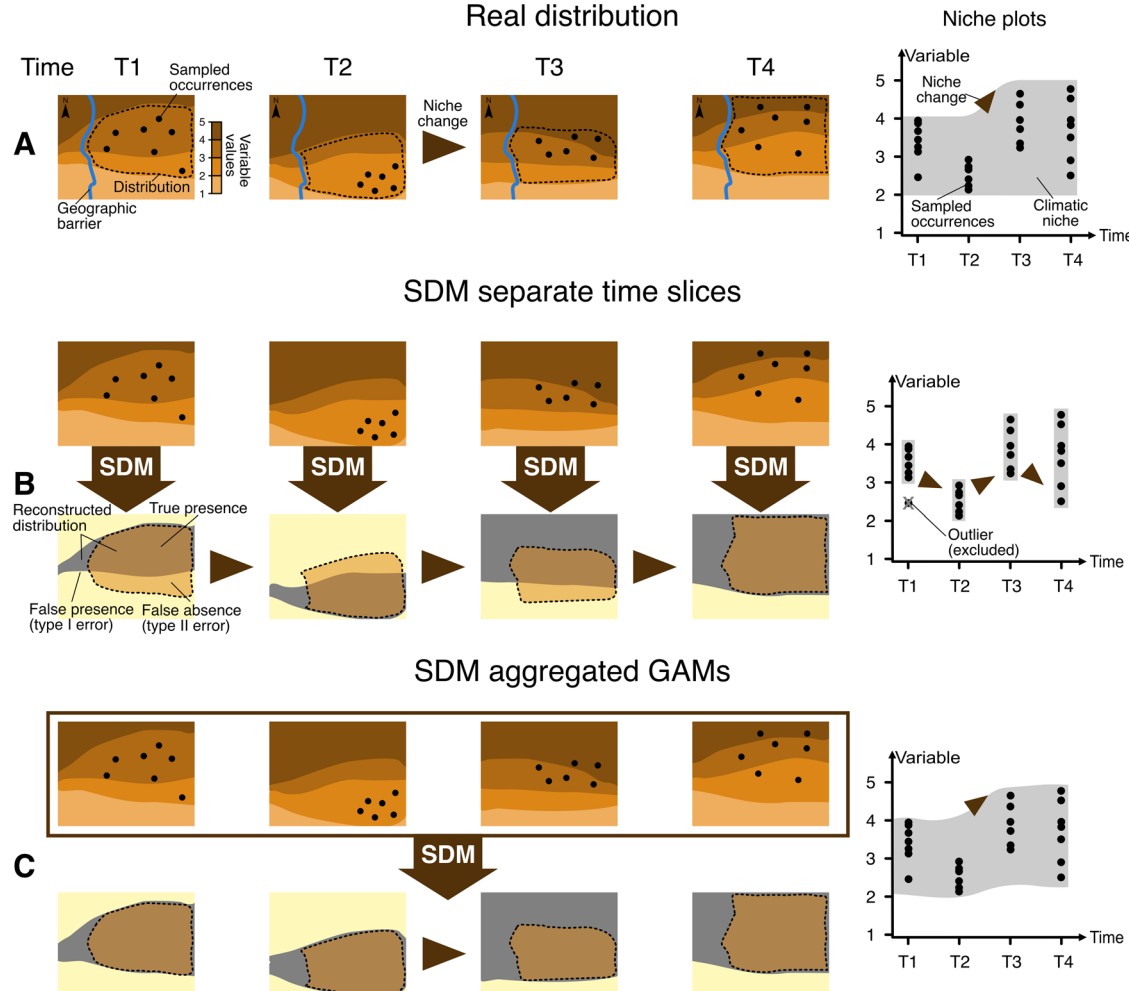

**Fig. 1 Schematic description of how different Species Distribution Modelling (SDM) approaches deal with diachronic data. A** Map of the actual distribution of a species (dotted line) through time in four different periods (T1–T4). Sampled occurrences are depicted as black points, and different values of the climatic variable are shown as different colour shades. On the right side, a niche plot shows how the niche (grey shading, with the *x*-axis representing a climatic variable of interest) varies through time (*y*-axis); and where occurrences are located within the niche space. Sections (**B**) and (**C**) show the different reconstructions of the distribution and niche changes for an SDM done separately over each time slice (**B**) and our aggregated GAM method presented in this work (**C**).

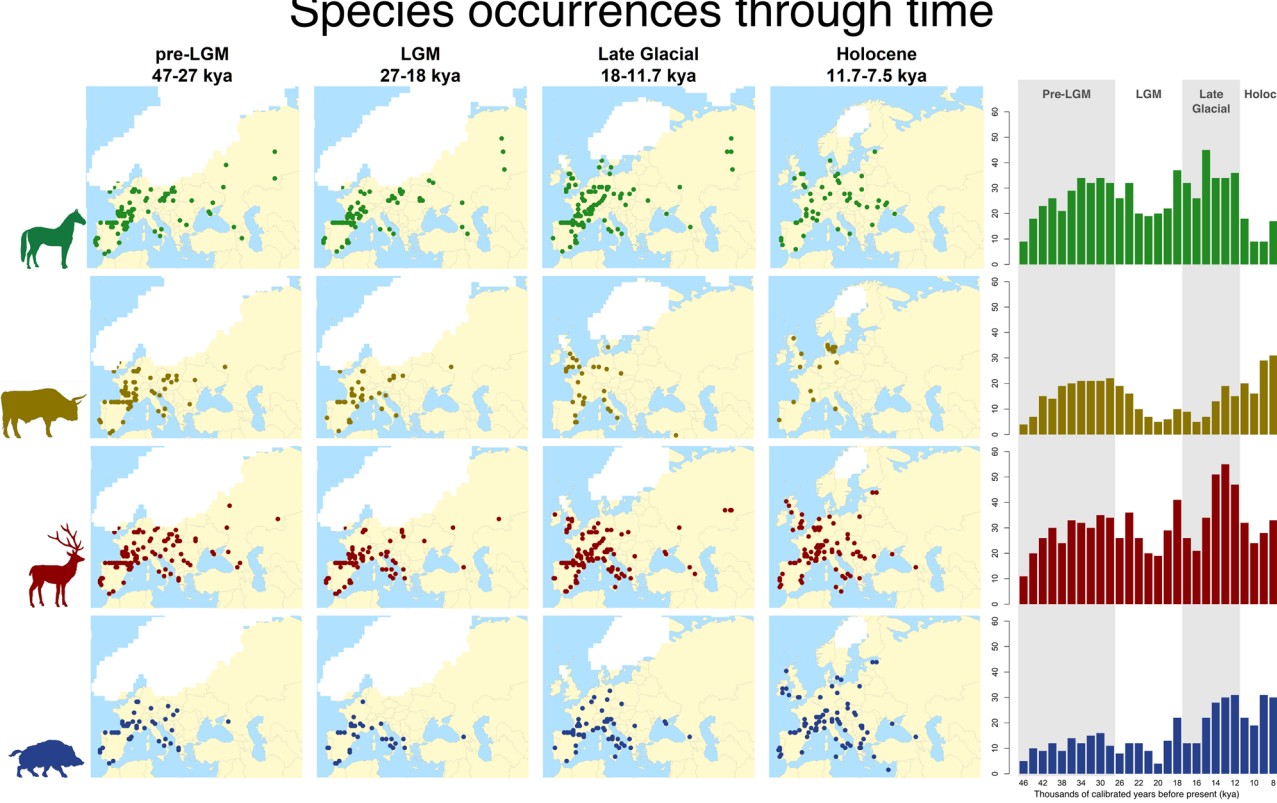

**Fig. 2 Geographic and temporal distribution of horse, aurochs, deer, and wild boar occurrences.** Dates are expressed in thousands of calibrated years before present (kya).

for a given time slice may lead to underestimating the niche space for that time slice (Fig. 1B, niche plots).

This undersampling, in turn, can have two significant consequences. The first is an underestimation of the potential distribution of the species over space in each time slice due to unsampled realized niche space because of sample bias. A possible solution is to aggregate the observations from different time slices before performing the analyses[16], which assumes the niche to be constant. The second consequence of analyzing each time slice separately, is that the sparse sampling may lead to overestimating their differences, identifying niche changes even when they have not occurred (Fig. 1B).

On the other hand, Species Distribution Modelling applied to animal movement analysis using tracking data has explicitly modelled changes in preference over time[17–19]. Following this approach, we use the Generalized Additive Model (GAM) framework to analyze all available data within a single model and test for niche changes over time. This aim can be achieved by fitting interactions (technically tensor products) between environmental variables and time, thus allowing the effect of those variables to change through time (Fig. 1C). Furthermore, this approach alleviates the patchy and limited sampling issue, as it can use the full-time series to test for changes in the use of a specific part of the environmental parameter space (Fig. 1C). In other words, we can consider whether a species was present before and after a particular time in that niche space and use that information to avoid forcing a niche change if there is only a temporary absence due to limited data.

Here, we use this approach on four European megafauna species that survived the sharp climatic changes that characterized the last part of the Pleistocene and the beginning of the Holocene: wild horses (*Equus ferus*), aurochs (*Bos primigenius*), wild boar (*Sus scrofa*), and red deer (*Cervus elaphus*, from now on "deer").

Horse and aurochs are adapted to more open environments (e.g., steppe, grassland, forest-steppe), while deer and wild boar tend to occupy more forested areas (with wild boars mostly favouring broadleaf forests). Moreover, horse and deer are better adapted to the cold within their respective habitat preferences, and deer have been more tolerant to different habitats[20–22]. This general picture is also confirmed in prehistoric times when the presence of large mammals in archaeological sites is compared to other proxies[23,24]. Furthermore, these species provide a range of representation in the archaeological record, thus sampling completeness, with horse and deer being very common and aurochs and wild boar much less frequent.

By analyzing their ecological niches and their variations through time, we can reconstruct if and how large herbivores with different habitat preferences reacted to the climatic fluctuations observed between 47 and 7.5 kya.

## Results

**Occurrences from the paleontological and archaeological record.** We collected direct or indirect radiocarbon dates (i.e. occurrences) from the literature and online databases for horse, aurochs, deer, and wild boar (see Methods for details). We removed any date older than 47 thousand calibrated years before present (kya), as there were too few data points for reliable analysis; or younger than 7.5 kya to avoid the confounding effect of domestication (Fig. 2).

We used the spatial coordinates and calibrated radiocarbon date for each occurrence to associate it with environmental variables. Palaeoclimatic reconstructions are available at a resolution of 0.5° for time slices of 1000 years up to 22 kya and 2000 years before that date. We focused on five variables which are relevant to large herbivores in Europe (see also[11]): maximum temperature of the warmest month (BIO5, from now on "maximum temperature");

# Variable importance

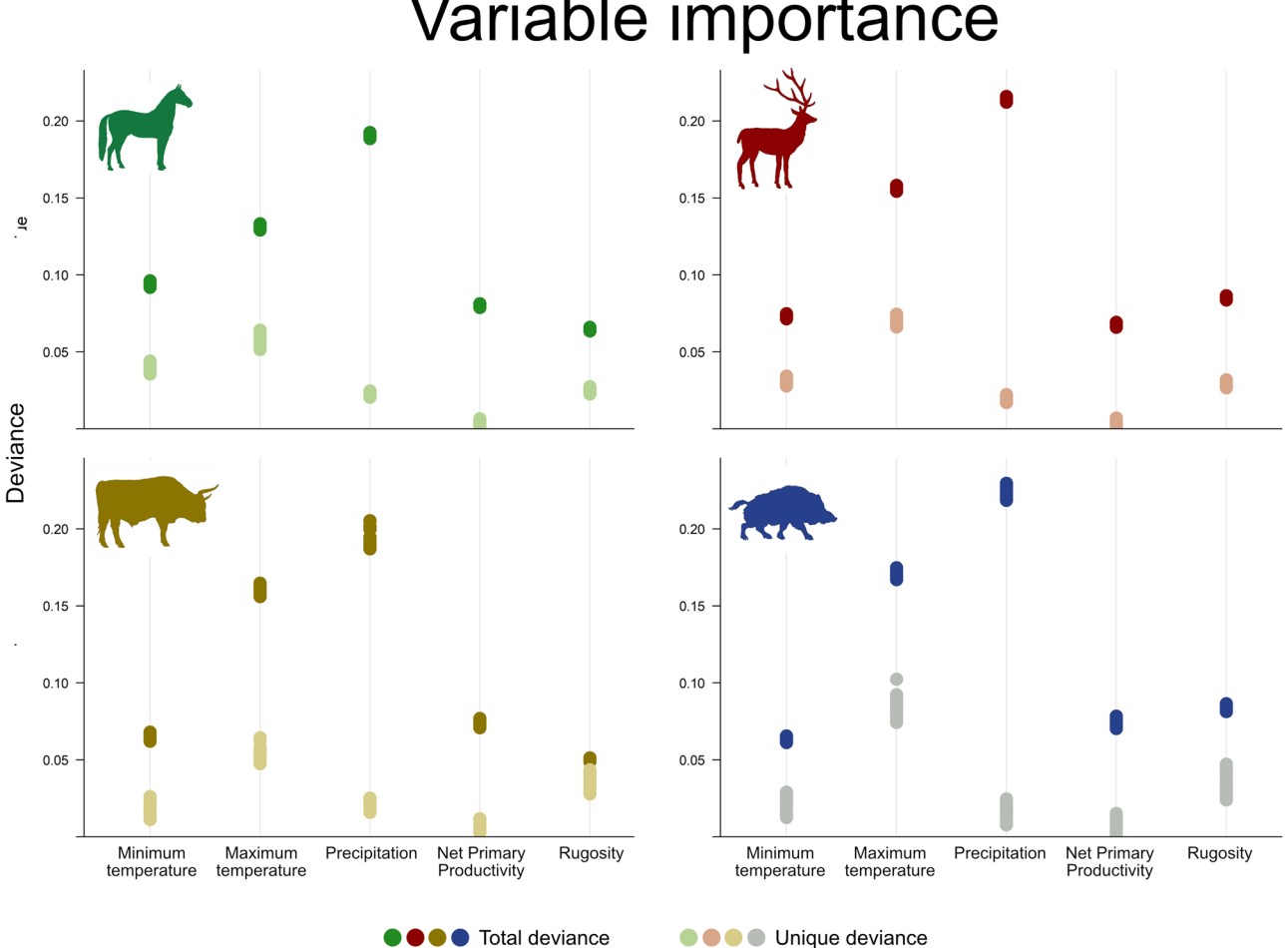

**Fig. 3 Variable importance.** Calculated both as the unique deviance explained by each variable (lighter points) and its total variance (darker points).

minimum temperature of the coldest month (BIO6, from now onwards "minimum temperature"); total annual precipitation (BIO12, from now on "precipitation"), net primary productivity (NPP), and a measure of rugosity to capture topography.

**Testing for niche change through time.** We fitted two models to each species, a simple GAM using only the environmental variables ("constant niche model") and a full model including the interactions with time ("changing niche model"). Since GAMs require a thorough sampling of the available environmental space, we randomly sampled 25 sets of background points (50 times the number of occurrences in each time slice) that could be paired with our occurrences used as presences. Thus, each model was fitted to 25 datasets ("repetitions"), including the occurrences and a different set of background points each. After checking for spatial autocorrelation (Supplementary Tables 1–4), we performed model selection on each dataset based on the Akaike Information Criterion (AIC). For all species, irrespective of the specific background set used, the changing niche model was better supported than the constant niche one (Supplementary Tables 5–8).

We then quantified the ability of the most supported model to successfully predict the occurrences through the Boyce Continuous Index (BCI)[25,26], using an acceptance threshold of 0.8 (Supplementary Table 9). For each species, the repetitions that exceeded such threshold were averaged in two ensembles (by mean and median), and the ensemble with higher BCI was used for all further analyses.

**Variable importance.** To better understand the environmental variables that underpinned the distribution of these four species, we calculated variable importance (Fig. 3), both as the total deviance (darker points) explained by each variable and its unique component (i.e. the deviance that was not explained by any other variable, lighter points). As expected from considering four temperate species, the patterns of variable importance were broadly consistent across species. Specifically, for all species, the variable with the highest unique deviance (lighter points in Fig. 3) was maximum temperature, followed by minimum temperature in colder-adapted species (horse and deer), and rugosity in ungulates with a bigger preference for warm areas (aurochs and wild boar). On the other hand, for all species, the total deviance (lighter points in Fig. 3) was highest for precipitation, followed by maximum temperature and rugosity in deer and wild boar.

**Potential distribution through time.** We used the best ensemble for each species (the one calculated using the mean in all cases) to project the potential distribution over each time slice. We then generated binary maps using the highest threshold, allowing us to recover 99% of each species' observations. To better visualize the changes in distribution, we considered four main climatic periods (pre-LGM, 47–27 kya; LGM, 27–18 kya; Late Glacial, 18–11.7 kya; and Holocene, 11.7– 7.5 kya), averaging the binary distributions for all the time slices within each period (Fig. 4). The binary distributions for each time slice are available in Supplementary Figs. 1–4.

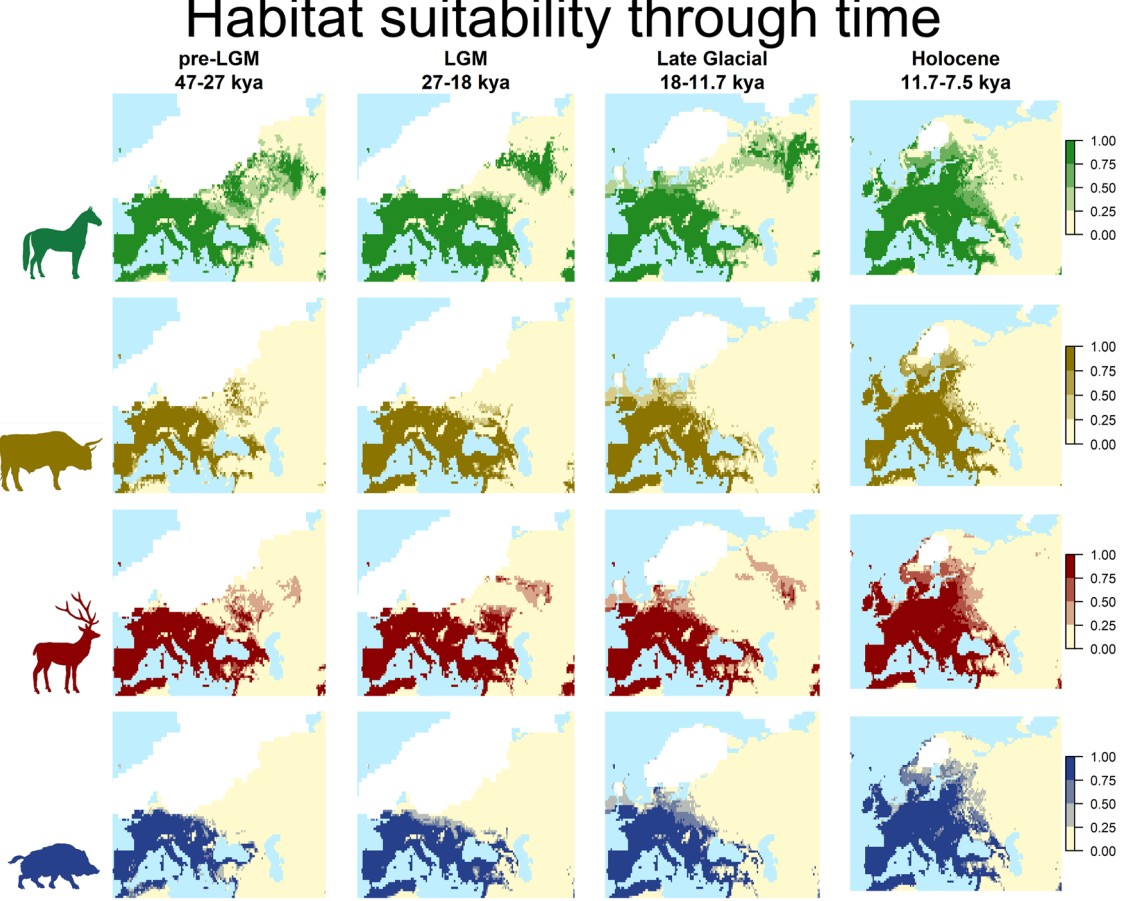

**Fig. 4 Potential distribution of the species for each climatic period.** Each map represents the mean distribution over time during each period, based on the best ensemble for each species. The maps show only cells defined as land in the whole period considered and the largest ice cover observed during the same time frame.

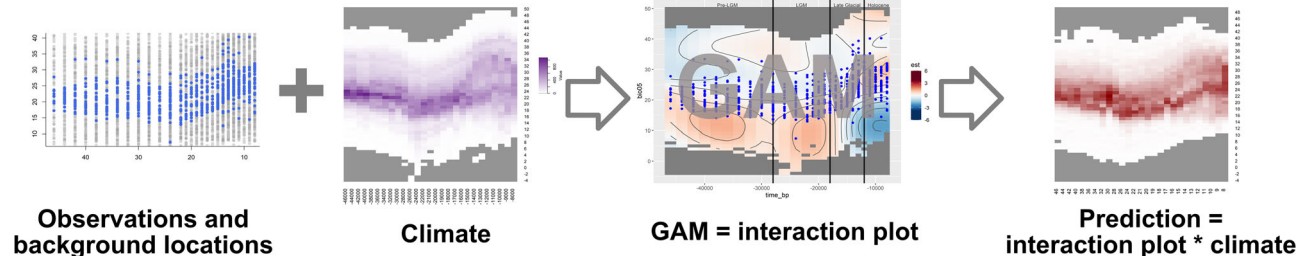

**Fig. 5 Schematic description of how the GAM capture the change through time in the effect of an environmental variable.** The example is based on the results for minimum temperature in deer.

During the pre-LGM and the LGM, wild boar and aurochs were restricted to Central, Western and Southern Europe, while horse and deer extended further towards the northeast. In Central Europe, for all four species, the northern limit of ranges seemed to be constrained mainly by the ice sheets. However, starting from the Late Glacial, the distributions of the four species became more similar, mainly due to the shrinking of horse and deer towards the west, reaching the same areas covered by wild boar and aurochs. At the same time, the retreat of the ice sheets allowed for an expansion, first northwards, and then eastwards during the Holocene.

**Niche change through time.** The easiest way to visualize the change in the niche through time is to use the interaction plots generated by the R package *gratia*[27] (Fig. 5).

The idea behind it is that the geographic distribution of a species over time (i.e., prediction) is affected by the availability of different environmental conditions (i.e., climate) and the ability of the species to use such conditions (i.e., its niche). The prediction of the species' distribution should be considered as the product of the relative abundance of different environmental conditions available in the area ("climate") and the GAM (the effect sizes shown in the "interaction plot"). The latter details the nature of the niche change over time where the smooths represent the effect of a variable on the predicted probability of occurrence. Because in a changing niche model, this effect changes through time, it can be visualized as a heatmap with time on the *x*-axis and the values of the variable of interest on the *y*-axis. In such a plot, the heatmap colour shows the effect on the probability of occurrence: red means an increased and blue a decreased probability compared to what would be expected

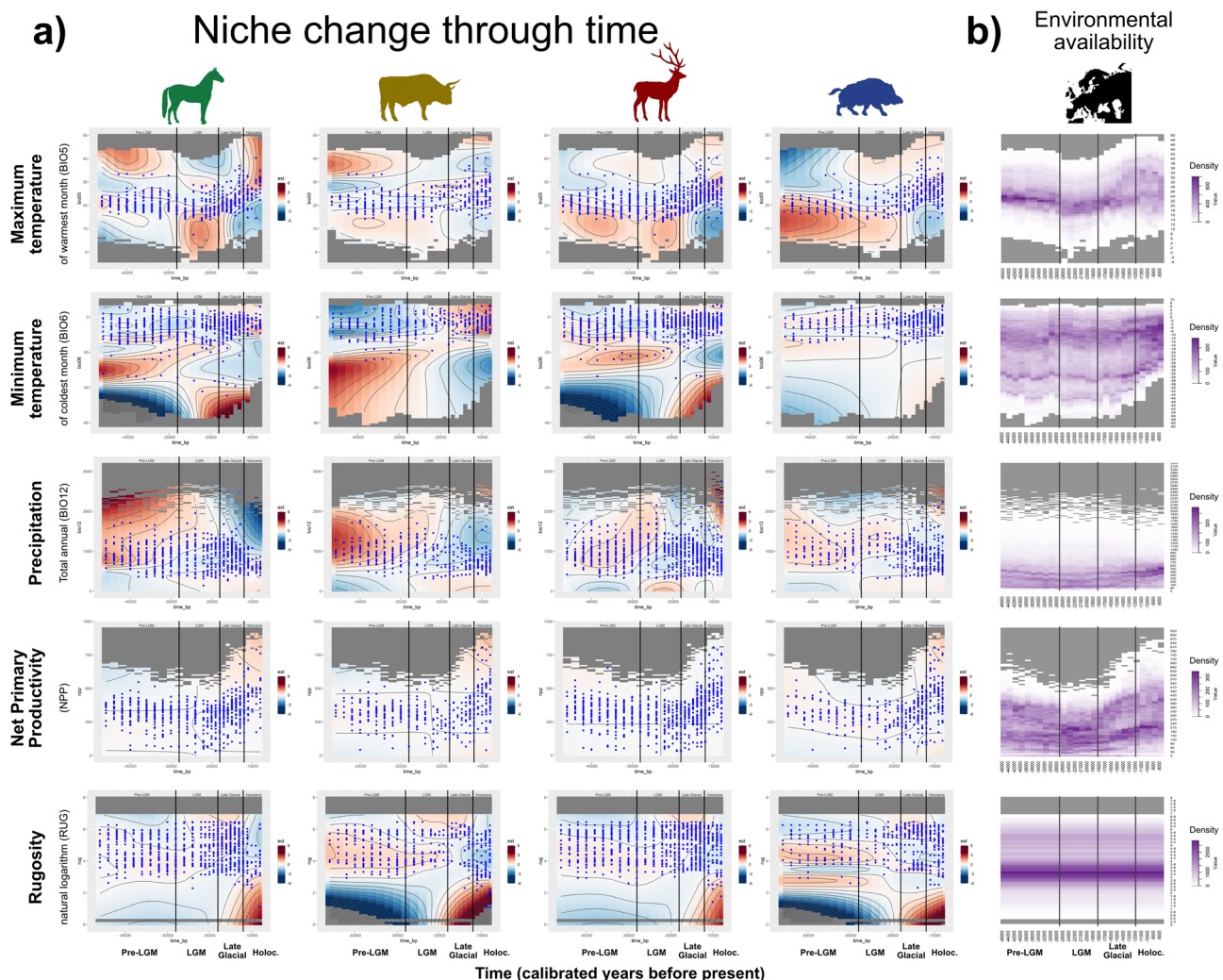

**Fig. 6 Interaction plots. a** Depiction of how the niche changed through time through heatmaps of the interaction between the environmental variables (**y**-axis) and time (**x**-axis): red represents areas in the variable space with higher relative preference, while blue is the opposite (see main text for a more detailed explanation). The colour scale is the same in all plots, black contours are isolines, and the blue dots show the distribution of the species observations for each time slice. **b** Environmental availability, i.e., density of each variable within Europe through time (the darker the shade, the higher the density). The variables are on the y-axis and time on the x-axis; the colour scales are the same across variables.

based on the distribution of values for a given variable, and black lines are isoclines.

If we had a variable with no effect (corresponding to a completely white smooth plot), we would still expect to see the species found most commonly in the most represented values for the environmental variable of interest. Availability is critical in determining the final distribution: a prevalent but less preferred environmental condition might still harbour a large portion of the range of a species compared to a favoured but scarce condition. Thus, to understand the impact of the changes in habitat suitability given by the smooths, we need to inspect them in conjunction with the occurrences and the changes in available environmental space. Finally, note that the variable with the largest change might not be the most important one in determining the overall distribution (see the earlier section on variable importance).

In Fig. 6a, we present the smooths for each species averaged over the fits to multiple background sets, in conjunction with presences (as blue dots) and the availability of environmental space through time (Fig. 6b). Since the colour scale is consistent among smooth plots, darker colours in Fig. 6a represent more considerable changes.

For example, the largest change for cold-adapted species (horse and deer) is observed in the effect of minimum temperature for colder areas. Between 47 and 16 kya, they are predicted to be utilized less intensely than what should be expected based on their availability (as shown by the large blue zone), but after that period, there is a shift towards using them more (occurrences appear in those areas and the plot shows them as dark red). The same can be observed for rugosity in warm-adapted species (aurochs and wild boar), while horse and deer appear to shift from high to low-rugosity areas only during the Holocene.

Precipitation shows bigger changes in open-habitat species: horse and aurochs. High-precipitation regions are over-occupied until the end of the LGM; after this period, they turn blue in the plots, as we can observe a reduction in the range of values covered by the observations.

For maximum temperature, forest species (deer and wild boar) tend to prefer lower values until the LGM and higher ones starting from the Late Glacial, while for the horse, the preference switch from high to low values at the end of the LGM, and then back to high values in the Holocene.

The whitish colour of the NPP plots shows that the distribution of the analyzed species closely follows NPP's environmental availability in the different periods, suggesting a quasi-random occupation of the space with respect to this specific variable.

## Discussion

The four species analyzed were predicted to occupy large parts of Western Eurasia, showing a degree of geographical overlap, coherent with them being common generalist species. However, at the same time, their ranges changed differently through time.

The distribution of horse and deer stretches towards the Ural Mountains until the LGM, while the other two species (better adapted to warmer climates) are restricted to Central and Western Europe (wild boar) or with a limited occupation of Eastern Europe (aurochs). This pattern likely reflects the observed niche changes with respect to minimum temperature. Not only are most differences located in the north-eastern steppes, which tend to be colder than the rest of ice-free Europe, but the pattern of preference changes through time (Fig. 6a) for minimum temperature closely matches the differences observed in their distributions, with horse and deer being more similar between them and aurochs behaving differently from everybody else.

The four potential distributions became much more similar from the Late Glacial onwards. However, an overlap in range does not necessarily imply identical habitat preferences: for example, the archaeological record from Holocene sites in Europe suggests that, even though our four species might have at time coexisted, horses (not adapted to forested areas) were relatively rare before domestication when the others were abundant[11,28–31].

Interestingly, the reconstructed range for our species did not shrink notably during the LGM, with their northern limits constrained mainly by the presence of the ice sheets (the same applies to the colder period before the LGM). This predicted distribution stretches further north than the southern areas suggested by the archaeological record[20,22,32,33] and genetic analyses (e.g.[34]). Their predicted ranges during the LGM encompass what are considered glacial refugia for temperate species during this period[32,35,36]: the Iberian, Italian and Balkan Peninsulas, Dordogne, and the Carpathians. However, our reconstructions cover a slightly larger area extending to regions not known to be inhabited by the analyzed species (e.g. most of Germany and the Carpathians for wild boar)[32].

There are several possible explanations for this discrepancy. First of all, when looking at Northern France and Germany, there is minimal archaeological evidence for the LGM[37]. Thus, a lack of observations might be a false negative due to inadequate sampling effort. The mismatch between predictions and observations could also be linked to such marginal regions being ecotones, i.e., transitional areas between two ecologically different zones. From an environmental point of view, they could be easily occupied by our species, but both the animal and plant communities would be a blend of the two neighbouring zones. This would cause biological interactions (e.g., competition, predation) that could lead to the species being less frequent than in their core areas and thus less observed in the archaeological record. This effect could be further compounded by differential predatory choices by human populations through space and time, a factor that may heavily influence the presence or absence of species in the archaeological assemblages. For instance, our previous SDM analyses on horse[11] showed that, although during the Holocene, a large part of Europe was still environmentally suitable for the species (which was apparently still widespread throughout the continent), it was instead mostly absent from the zooarchaeological record, with a presence in faunal assemblages usually less than 3 % of the identified remains. A further complication is that our climatic reconstructions are inevitably coarse (each cell is approximately 100 km wide); whilst the average environmental conditions might be suitable for the species, heterogeneities in microclimate might lead to habitat fragmentation that could greatly reduce species density or even preclude the viability of populations. A similar issue was discussed in[11,31]: when horse remains were found in areas where the macroclimate suggested a forested environment, they were shown to live in open areas based on microhabitat reconstructions based on the faunal assemblages at archaeological sites.

Despite the patchy and limited sampling of archaeozoological remains, we were able to capture the differential adaptation to mountainous areas. The higher-altitude areas in the Alps and the Carpathian mountains were considered unsuitable for deer and wild boar until the Holocene, corresponding with the expansion of forests in these areas. Mountainous areas remained unsuitable for horse and aurochs throughout the whole study period. However, it must be noted that our climatic reconstructions do not include the alpine ice sheet, which would have acted as a physical and hence genetic barrier for our species as it is, for example, suggested for aurochs[38,39]. On the other hand, the higher-altitude areas in the Caucasus show a different pattern, as they become more unsuitable for aurochs and wild boar starting from the LGM and for horse and deer in the Holocene. We would caution against overinterpreting the lack of signal from other mountainous regions: whilst it might derive from their lower altitudes (e.g., the southern Balkans), the inevitably coarse scale of our reconstructions might have prevented us from detecting the effect of mountains with a smaller footprint such as the Pyrenees (which were shown to have acted as a geographic barrier for horses[11,40,41]). It is of interest that the area covered by the sea of Azov, despite not being excluded from land in the climatic reconstructions[42] is still considered unsuitable for all of our species during the whole period. This suggests that both the reconstructions and the modelling are robust.

Using our method, we showed that all four species changed their niche during or just after the LGM, with such changes mainly being driven by minimum temperature and rugosity. However, we note that, even though the models did pass many quality checks, those two variables are difficult to disentangle, as areas of high rugosity tend to have colder temperatures.

Our analysis only considers bioclimatic factors and topography, and dispersal and biotic interactions are not evaluated. Besides potentially missing specific changes in the niches[43], it is also important to note that observed changes could be indirect effects of some of these unmeasured factors. The synchrony of niche change in these species suggests such an external factor. For example, the onset of the LGM triggered significant shifts and a dramatic reduction of the ranges of many herbivores and their predators, potentially reshaping animal communities[44–46]; thus, the observed changes in niche might result from the removal of competitors and predators. Moreover, another important phenomenon that might have influenced these ecological dynamics is the drastic demographic decrease and the consequent genetic bottleneck suffered by human populations[47] as well as changes in their hunting technology which looks to have been, at some level, influenced by ecological constrains[48].

Our analyses show that niche changes can occur within time frames in the order of tens of thousands of years. From an evolutionary perspective, identifying changes in the realized niche is a crucial starting point for testing whether they are linked to specific adaptations (e.g., using ancient DNA data) or shifts within a large fundamental niche (e.g., as a response to a change in biotic interactions). These aspects could be highly significant when planning conservation or restoration efforts. In particular, if the changes observed in our data are linked to changes in community composition, this requires special attention for future projections, given the extent to which species go extinct[49,50].

**Table 1 Number of observations for each species.**

|  | Horse | Aurochs | Deer | Wild boar |
|---|---|---|---|---|
| Original dataset | 1725 | 892 | 1903 | 870 |
| Collapsed datasets | 694 | 401 | 823 | 430 |

Original dataset: number of observations collected; Collapsed dataset: number of observations retained after keeping only one presence per time slice and grid cell.

## Materials and methods

**Materials**. We collected from the literature and available databases a dataset of radiocarbon dates from Europe (West of 60°E and North of 37°N) either obtained from remains of the four analyzed species or from archaeological layers where they have been observed. However, we only considered observations dated between 7500 and 47,000 cal BP: their scarcity before this period may bias the GAMs, and after it, domesticated cattle, pigs and (later) horses arrived in Europe, making it difficult to differentiate them from their wild forms.

We excluded any record fitting one or more of the following conditions: unreliable; not in accord with the expected chronology of their archaeological layer; without a reported standard error; available only as *terminus ante/post quem*.

All dates were calibrated with OxCal[5] version 4.4 using the IntCal20 curve[51], and we further excluded any record for which calibration resulted in an error, resulting in the number of points presented in Table 1 as "Original dataset" (available at the link https://doi.org/10.6084/m9.figshare.20510364).

SDMs based on GAMs need presence/background data, not frequencies; moreover, multiple observations (i.e., presence in different archaeological layers) from the same site and time slice are likely to introduce stronger sample biases linked to chrono-geographically differential sampling efforts. For this reason, we collapsed our observations by keeping only one point per grid cell per time slice for each species, leaving the number of observations reported in Table 1 as "Collapsed datasets", used for all the analyses presented in this work.

To perform all analyses, we used the R package *pastclim* v. 1.0[42] to couple each observation from the collapsed datasets to paleoclimatic reconstructions published in[8] by setting dataset = "Beyer2020". These are based on the Hadley CM3 model, include 14 different bioclimatic variables at a spatial resolution of 0.5°, and are available for the whole world every 1000 years until 22 kya and every 2000 years before that date (referred to in the manuscript as "time slices"). Specifically, each observation was associated with the relevant bioclimatic reconstruction based on its average age and spatial coordinates.

As already mentioned, the four species analyzed show different preferences regarding temperature, habitat, and altitude. Therefore, for the Species Distribution Modelling, we choose five environmental variables that should be able to capture such differences: two measures of temperature (BIO5, maximum temperature of the warmest month, and BIO6, minimum temperature of the coldest month); two variables to help capture habitat differentiation (BIO12, total annual precipitation, and Net Primary Productivity, NPP), and one measure of topography (rugosity[42]).

High collinearity can be problematic in SDMs; we confirmed that all our variables had a correlation below 0.7, a threshold commonly adopted for this kind of analysis[52,53].

Whilst the GAMs predicted all time points; we visualized our results by creating an average estimate for the following periods: pre-LGM (from the beginning of the time range analyzed, i.e., 47 kya to 27 kya), LGM (from 27 to 18 kya), Late Glacial (from 18 to 11.7 kya), Holocene (from 11.7 kya to the end of the time range analyzed, i.e., 7.5 kya).

**Methods**. We generated 25 sets of background points for each species to adequately represent the existing climatic space in our SDMs. Each set was generated by sampling, for each observation, 50 random locations matched by time. This resulted in $n = 25$ datasets ("repetitions") of background points and presences (observations) for each species, which we used to repeat our analyses to account for the stochastic sampling of the background. For each dataset, we used GAMs to fit two possible models: a "constant niche" model, which included only the environmental variables as covariates, and a "changing niche" model, that also included interactions of each environmental variable with time (fitted as tensor products).

In GAMs, the effect of a given continuous predictor on the response variable (in our case, the logit transformed probability of a presence) is represented by a smooth function; this smooth function can be linear or non-linear and can become highly complex in shape depending on the number of knots selected by the GAM fitting algorithm. The interaction between two covariates is modelled by tensor products[54]; this approach is equivalent to an interaction term in a linear model but with the added complexity of the smooth function. In our models, we confine tensor products to the interaction between an environmental variable and time; a simple way to think about such a tensor product is that it allows the smooth representation of the relationship between the variable and the probability of a presence to change progressively over time.

GAMs were fitted using the *mgcv* package in R[54] using thin plate regression splines (TPNR; bs = "tp", default in *mgcv*) for environmental variables and their tensor products with time in the "niche changing" models. The GAM algorithm automatically selects the complexity of the smooth most appropriate to the data that are being fitted; as GAM can have issues with overfitting, we added an additional penalty against overly complex smooths (gamma = 1.4) and used Restricted Maximum Likelihood (REML = TRUE), as recommended by[54]. It is possible that even with these settings, the complexity of the smooth is not sufficient; we used *mgcv::gam.check()* to check this, and increased the basis dimension of the smooth, k, to make sure that k-1 was larger than the estimated degrees of freedom (edf). We found the best maximum thresholds for k to be 16 for bio06 and 10 for all other variables.

We checked for non-linear correlation among variables using the *mgcv::collinearity* function and checked the values of estimated concurvity. All estimates were below the threshold of 0.8 in all models, runs and variables except for a few instances for time (Supplementary Figs. 5–8). We consider this not to be worrying: this is most likely a result of sample bias, and GAM is known to be robust to correlation/concurvity[55,56].

We verified the model assumptions by inspecting the residuals using the R package DHARMa[57]. Standard tests for deviations from the expected distribution and dispersion were non-significant for all repetitions for all species, as were the tests for outliers. Furthermore, we tested for spatial autocorrelation among residuals by computing Moran's I; all tests were either non-significant or, when significance was detected, the estimate of Moran's I was very close to zero, revealing a trivial deviation from the assumptions which should not impact the results (Supplementary Tables 1–4).

We performed model choice (Supplementary Tables 5–8) by comparing the constant- and changing-niche models for each combination of species and repetition using the Akaike Information Criterion (AIC). AIC strongly supported the changing-niche model in all species and repetitions, an inference supported by the higher Nagelkerke $R^2$ and expected deviance for those models than for the constant-niche ones (Supplementary Tables 5–8).

The model fit for each of the changing niche GAMs was evaluated with the Boyce Continuous Index[25,26], designed to be used with presence-only data[58,59]. We set a threshold of Pearson's correlation coefficient > 0.8 to define acceptable models[25] (Supplementary Table 9).

The relative importance of each environmental variable was quantified for all the models above the BCI threshold of 0.8 in two different ways. Firstly, we computed the total deviance explained by each variable by simply fitting a GAM with only that variable. We then estimated the unique deviance explained by each variable by comparing the full model with one for which that variable was excluded (i.e., we computed the explained deviance lost by dropping that predictor. The difference between the two values represents the deviance explained by a variable which can also be accounted for by other variables (i.e., the deviance in common with other variables).

To achieve more robust predictions[60], we averaged in two different ensembles the repetitions for the changing niche GAMs with BCI > 0.8: by mean and median. This step is intended to reduce the weight of models that are highly sensitive to the random sampling of the background[60]. Then, for each species, we selected the ensemble (either based on mean or median) with the higher BCI as the most supported and used it to perform all further analyses.

The effect of different variables through time was visualized by plotting the interactions of the GAMs. For each model with a BCI > 0.8, we used the R package *gratia*[27] to generate a surface with time as the *x*-axis, the environmental variable as the *y*-axis, and the effect size as the *z*-axis (visualized as colour shades). We then plotted the mean surface for each species, which captures the signal consistent across all randomized background sets.

To visualize the prediction for each species, we then transformed the predicted probabilities of occurrence from the ensemble into binary presence/absences by using the threshold needed to get a minimum predicted area encompassing 99% of our presences (function *ecospat.mpa()* from the *ecospat* R package[61]). The binary predictions were then visualized using the mean over the time steps within each major climatic period.

**Reporting summary**. Further information on research design is available in the Nature Research Reporting Summary linked to this article.

## Data availability

All data analyzed during this study can be accessed at the link https://doi.org/10.6084/m9.figshare.20510364.

## Code availability

The complete code used to perform the analyses presented in this study can be accessed at the link https://doi.org/10.6084/m9.figshare.20510364.

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

## Acknowledgements

M.L. and A.M. have been funded by the ERC Consolidator Grant 647787 "LocalAdaptation" and the Leverhulme Research Grant RPG-2020-317. In addition, the authors thank Elisa Anna Fano and Mario Coiro for their helpful comments on the manuscript and Robert Sommer for the valuable discussion of the results. Finally, the authors are very grateful to the three reviewers for their constructive comments that greatly improved the quality of the manuscript.

## Author contributions

M.L. and A.M. designed the study with inputs from F.B. and P.B. M.L., F.B. and P.B. collected the data and curated the dataset. M.L. performed the analyses and designed the figures under the supervision of A.M. M.L. and A.M. provided the interpretation with inputs from F.B. and P.B. M.L. and A.M. wrote the original draft of this manuscript with inputs from F.B. and P.B. All authors commented on and edited the final version of the manuscript.

## Competing interests

The authors declare no competing interests.
