## [Peer Review File · Communications Biology]

Reviewers' comments:

Reviewer #1 (Remarks to the Author):

General comment

Dr Leonardi et al. present here a paper I would have loved to write myself. It is well and clearly written, the narrative is compelling, the figures are nicely done, the interesting methods (spatio-temporal GAM) are rarely seen in the SDM literature, and the complete supplementary material, including the full datasets and the code used for the analyses, almost allows full reproducibility (the environmental layers are missing though). I have enjoyed reading a good deal, and want to commend the good work the authors have put into developing this paper.

I only have substantial but easy to address comments about how the potential advantages of the models are presented (lines 327 or 330, for example), and how they are evaluated (line 392). I think that the spatio-temporal approach based on GAMs presented in the paper is very interesting and innovative, and has the potential to move the field of palaeo-SDMs forward. However, I feel that the ability of this method to deal with undersampling, or to be "the best possible approximation of the fundamental niche of a species" is either overstated, or at least, not demonstrated, and the reader has to take the author's claims at face value. Please, be aware that this comment might be just the result of my own lack of expertise in the spatio-temporal approach presented by the authors, but somehow I feel that either providing proof or tuning down these claims might be the alternative ways to go.

About model evaluation, the authors use the Boyce Continuous Index, but do not show the goodness of fit of the presented GAMs, either via R-squared or explained deviance, ideally both, nor do they show residuals tests in the supplementary material, or comment on the potential effects of spatial or temporal autocorrelation on the model results. Somehow, I feel that the quality of the GAM models is hidden behind the Boyce Continuous Index, and I believe this aspect of the paper can be improved.

Below I have several specific comments for the authors.

Line 42: The word "defined" appears twice in a row, and a couple of commas are missing. I suggest rewording the sentence as follows:

"The ecological niche of a species, as defined by Hutchinson in 1957 (3), can be separated into two fractions, the fundamental and the realized niche. The fundamental niche is the range of biotic and abiotic conditions in which a species can survive and reproduce."

Line 45: I think the sentence "However, certain conditions that would, in theory, be suitable, might be realized in the physical space that can be occupied by the species at a given time..." is either hard to follow, or misconstructured. I'd love to suggest an alternative, but I am not sure about what the authors mean by it. Probably, simplifying and splitting it in two might help.

Line 52: I suggest replacing "climatic variables" with "environmental variables", since topographic and other types of variables can be used to fit SDMs.

Line 94: there is a comma missing after "tricky aspect".

Line 95: I suggest replacing "within the time frame" with "over time scales between ...".

Line 99: The use of "time frame" is becoming reiterative, I think that "in a very short time" works well instead.

Line 100: I suggest replacing "doubtful" with "unclear".

Line 106: I think that "we adapt it to look at diachronic data" could be replaced with "we adapt it to work with diachronic data".

Line 166: I think this sounds a bit better like this "..., the constant niche model showed a poor fit with the data (...).

Line 168: As above, I would replace "was able to describe the data" with something along the lines of "showed an adequate fit".

Lines 176 to 183, but the comment is valid for other places as well. I find weird the use of the past tense to describe the distributions shown by the models, such as "The auroch tended to occupy" (but please, take in mind that I am a non-native English speaker, so I might be totally off). In my opinion, the models are fitted, and in these models, the "auroch tends to occupy...", so it sounds better with the present tense. Of course, this is an irrelevant stylistic comment, so the authors have nothing to do here unless they want to.

Line 194: when the authors say "recover 99% of our data" they are likely referring to "recover 99% of the presence records of each species", right? If that's the case, please, clarify it, as "our data" sounds a bit unspecific here.

Line 220: The sentence "..., represented by the unique deviance explained by each of them." sounds quite unspecific as well. I know the authors are talking about predictors/variables/covariates, so maybe it would make sense to use whatever word they prefer rather than "them".

Line 220: how is the "unique deviance explained" computed for each predictor? I found no mention in the methods (but I haven't read the supplementary material yet).

Line 308: replace "dispersals" with "dispersal".

Line 322: replace "strength" with "strengths".

Line 322: I tend to disagree with the statement "By using the archaeozoological record, it overcomes the bias given by contemporary ...". Human pressure was already there since at least 300ky, and was likely shaping the distribution and population dynamics of megafauna at that time frame. It is totally true though that the magnitude of the human impact has increased exponentially, but to claim that presence records between 40 and 7.5ky are not affected by human pressure might not be totally safe, more so when the potential impact of human presence, that could be inferred from the archaeological record, has not been assessed here. My point really is "does using the archaeozoological record really help sdms overcome the bias given by contemporary human impact?", or in other words, if the sentence is assumed as truth, does the palaeo-sdm field benefit from such an assumption, or are we just disregarding a potential source of bias we actually do have data to address?

Line 327: When the authors say "Our method, by taking into account all occurrences through time, is less sensitive to this problem...", why is the method less sensitive to undersampling? I cannot see why a lack of presence records during a time-slice would not produce in the GAM an apparent reduction in niche breadth. I agree that the spatio-temporal SDMs fitted in this paper are quite interesting, but to affirm that they can overcome a limitation others method cannot without a proper explanation isn't optimal.

Line 330: here the authors affirm that "our approach provides the best possible approximation of the fundamental niche of a species", how so? For example, using true absences (and I understand how difficult is it to extract absences from archaeological contexts) would be a better approach to define niche boundaries than pseudo-absences. My point is that every method has its limitations, and turning a gam into a spatio-temporal sdm doesn't necessarily overcome the limitations sdms already have.

Line 392: why are the GAM models evaluated only with the Boyce Continuous Index, but not with their own measures of goodness of fit. The adjusted R squared and the explained deviance should be provided as well. Also, how are the model residuals? Are they normal and spatially

autocorrelated? I think all these details would provide the reader with a deeper insight into how good the fitted models are, and in my opinion, the Boyce Continuous Index is not enough in this case.

Figure 5: How is the variable importance computed? I have seen it in the code, but a few words about it either in the methods section or the figure caption would go a long way.

Thanks for such a great paper, I am looking forward to seeing it published.

Blas M. Benito

Reviewer #2 (Remarks to the Author):

Leonardi et al. use a multi-temporal calibration approach to ecological niche modelling for four European ungulates from the pre-LGM (40kya) to the Holocene (8kya) in order to assess potential niche shifts across this time period. They find that red deer and wild boar did not change niches, as both species have relatively broad climatic niches, while horse and aurochs both expanded their more restricted niches after the LGM and the authors discuss possible reasons, including loss of megafaunal grazing competitors.

In general this is a nice paper, tightly focused on a complementary set of ungulates and making good use of the long-term fossil record to highlight species' long-term niches using a range of statistical approaches.

However, there are a few areas where more detail or clarification is required, in particular the methods which I have outlined below in more detail, as it's difficult to assess your results/discussion without fully understanding the process you've gone through with the data.

General comments

There are a few places where the semantics around niches gets confusing. In this paper you are really only discussing a species' (bio)climatic niche, or perhaps to be straightforward, its climate envelope. A huge number of different biotic and abiotic factors go towards defining realised niches, not just climate. It may help to clarify by using the term 'climatic envelope' or 'climatic niche' throughout.

Line 46: should this crucially read 'might not be'?

Line 57-59: a little confusing, do you mean a portion of their fundamental niche here? If human activities are restricting species distributions then is that not a constraint on the fundamental niche and the resultant distribution is the realised niche at that point in time? It may help to change 'realised niche in terms of climatic variables' to 'climatic niche'?

Line 61-64: Again, perhaps I am misunderstanding but by using palaeorecords are you not helping to more widely define the climate envelope (part of its fundamental niche) of a species rather than its realised niche? Then by constraining the projections using the available climate at each different time point you are plotting the potential realised (climatic) niche at each different time point. If the projections are at a mismatch with fossil records, or other ground-truthing data, then you may then be able to surmise that factors other than bioclimatic variables are constraining the realised niche (as per your discussion)?

Line 102-3: 'niche variation through time' – again, you must be referring to realised niche here.

Line 118-121: can you provide evidence/references as to what the different species' fidelity to 'open' vs 'closed' habitat is based on (dentition, fossil records)? These assumptions underpin most of your interpretation of the data which is interesting given your approach is about not assuming static niches of species. Your results seem to suggest all four species inhabited a variety of

habitats as their distributions are very wide!

I found Figure 3 and the associated analysis a little confusing to interpret and not sure I followed your results in lines 176-7!

Line 199: it looks from Figure 4 like most species expanded or maintained their ranges into the LGM? Do you have a quantification of range that supports your assertion that ranges contracted?

Lines 199-204: northern limits of ranges seem to only constrained by the ice sheets in most cases?

Line 214: I cannot find Figure 5A so cannot assess this part of the results.

Lines 232-233: Can you explain this? I suspect it's because productivity of vegetation (herbs) is important for grazers, whereas precipitation influences shrub/forest growth?

Line 240-242: But from your maps it looks like the aurochs does inhabit the Caucasus pre-LGM and LGM, and the horse in LGM? There is a relatively good Holocene fossil record for these species in the Caucasus too.

Line 265-6: Could this be due to confusion between domestic and wild horse i.e. potentially a lot of wild horse not identified or down as *Equus* sp?

Line 289: 'vegetal' = plant?

Methods

Much more detail is needed in the methods around primary data collection, as what is fed into the models will be highly influential to model performance and predictions. It would also be very hard to replicate this study without more information on data collection.

Line 343-344: 'whilst for the other three species we collected direct and indirect radiocarbon dates from the literature' – far more information is needed on how radiocarbon dates were collected, as online databases notoriously have a large amount of unreliable data that has been uploaded by multiple people and not checked. For example, which online databases were used? What were the criteria for selecting 'good' i.e. reliable dates? What were the criteria for selecting indirect dates? There is huge potential in indirect dates for including data that do not relate to your focal species without clearly thought out auditing criteria.

Line 347: 'all dates were...then classified into time slices'. Do these time slices refer to the final four time periods used to present the data in the paper, or the 1000-2000 year time slices provided by the bioclimatic data that are presented in the SI?

How were data classified into the time slices e.g. did you use the mid-point of each calibrated radiocarbon date; the weighted mean; did you include a record in every time slice that it fell?

Line 351: Describing the 'cleaning' of this dataset here is very confusing as it turns out to be the 'raw' dataset, and not the one you go on to call 'cleaned' (line 360)! Could the language here be adjusted for clarification?

Line 351-353: Was this the only cleaning of data that occurred? Did you double check species IDs, that dates were from a reliable lab, that a date wasn't old and/or hadn't been superseded by re-dating, did you check references etc? There is a large amount of 'dodgy' dates in online databases so it's really important to demonstrate at this stage that you've taken steps to ensure your data is as reliable as possible.

Line 359: Why did you collapse the data (I assume to reduce spatial bias but it would be helpful to say why)? Did you do this for each species?

Lines 361-367: Why were these variables chosen i.e. what was the biological reasoning? And again, how were they linked temporally to calibrated radiocarbon dates?

Line 385-6: Can you clarify how 'time' was used as a variable as I am not familiar with tensor products and cannot visualise how it was used in the model?

Reviewers' comments:

*Answers are written in blue and preceded by an asterisk

Reviewer #1 (Remarks to the Author):

General comment

Dr Leonardi et al. present here a paper I would have loved to write myself. It is well and clearly written, the narrative is compelling, the figures are nicely done, the interesting methods (spatio-temporal GAM) are rarely seen in the SDM literature, and the complete supplementary material, including the full datasets and the code used for the analyses, almost allows full reproducibility (the environmental layers are missing though). I have enjoyed reading a good deal, and want to commend the good work the authors have put into developing this paper.

*Answer: we thank the reviewer for their kind words

I only have substantial but easy to address comments about how the potential advantages of the models are presented (lines 327 or 330, for example), and how they are evaluated (line 392). I think that the spatio-temporal approach based on GAMs presented in the paper is very interesting and innovative, and has the potential to move the field of palaeo-SDMs forward. However, I feel that the ability of this method to deal with undersampling, or to be "the best possible approximation of the fundamental niche of a species" is either overstated, or at least, not demonstrated, and the reader has to take the author's claims at face value. Please, be aware that this comment might be just the result of my own lack of expertise in the spatio-temporal approach presented by the authors, but somehow I feel that either providing proof or tuning down these claims might be the alternative ways to go.

*Answer: both reviewers commented on this and similar statements. After careful consideration we decided to remove them in order to provide a clearer manuscript, more focussed on the analyses and the results than on the methods.

About model evaluation, the authors use the Boyce Continuous Index, but do not show the goodness of fit of the presented GAMs, either via R-squared or explained deviance, ideally both, nor do they show residuals tests in the supplementary material, or comment on the potential effects of spatial or temporal autocorrelation on the model results. Somehow, I feel that the quality of the GAM models is hidden behind the Boyce Continuous Index, and I believe this aspect of the paper can be improved.

*Answer: we have modified the methods to address this comment. We now perform model choice with AIC, which is the standard test to do it, and also considered explained deviance and Nagelkerke R^2 (that are always in accord with AIC) (Supplementary tables 1-4). Moreover, we now included in the analyses also measures of the spatial autocorrelation in the model residuals (Moran's I, supplementary tables 5-8) and performed standard tests on the residuals (discussed in the text but not shown; the associated code is available in the supplementary material).

Below I have several specific comments for the authors.

*Answer: we have heavily modified the text, so some of the portions to which the comments refer may not be present anymore. For your references, the reviewer's comments have been

added next to the associated (even if deleted) text in the manuscript file with marked revisions.

Line 42: The word "defined" appears twice in a row, and a couple of commas are missing. I suggest rewording the sentence as follows:

"The ecological niche of a species, as defined by Hutchinson in 1957 (3), can be separated into two fractions, the fundamental and the realized niche. The fundamental niche is the range of biotic and abiotic conditions in which a species can survive and reproduce."

*Answer: the whole paragraph has been removed

Line 45: I think the sentence "However, certain conditions that would, in theory, be suitable, might be realized in the physical space that can be occupied by the species at a given time..." is either hard to follow, or misconstrued. I'd love to suggest an alternative, but I am not sure about what the authors mean by it. Probably, simplifying and splitting it in two might help.

* Answer: the whole paragraph has been removed

Line 52: I suggest replacing "climatic variables" with "environmental variables", since topographic and other types of variables can be used to fit SDMs.

*Answer: addressed as suggested

Line 94: there is a comma missing after "tricky aspect".

* Answer: the whole paragraph has been removed

Line 95: I suggest replacing "within the time frame" with "over time scales between ...".

* Answer: the whole paragraph has been removed

Line 99: The use of "time frame" is becoming reiterative, I think that "in a very short time" works well instead.

* Answer: the whole paragraph has been removed

Line 100: I suggest replacing "doubtful" with "unclear".

* Answer: the whole paragraph has been removed

Line 106: I think that "we adapt it to look at diachronic data" could be replaced with "we adapt it to work with diachronic data".

*Answer: the sentence has been changed as

"On the other hand, Species Distribution Modelling applied to animal movement analysis using tracking data has explicitly modelled changes in preference over time 17–19. Following this approach, we use the Generalised Additive Model (GAM) framework to analyse all available data within a single model and to test for niche changes over time."

Line 166: I think this sounds a bit better like this "..., the constant niche model showed a poor fit with the data (...).

*Answer: the whole paragraph has been removed

Line 168: As above, I would replace "was able to describe the data" with something along the lines of "showed an adequate fit".

*Answer: the whole paragraph has been removed

Lines 176 to 183, but the comment is valid for other places as well. I find weird the use of the past tense to describe the distributions shown by the models, such as "The auroch tended to occupy" (but please, take in mind that I am a non-native English speaker, so I might be totally off). In my opinion, the models are fitted, and in these models, the "auroch tends to occupy...", so it sounds better with the present tense. Of course, this is an irrelevant stylistic comment, so the authors have nothing to do here unless they want to.

*Answer: we modified the text as suggested

Line 194: when the authors say "recover 99% of our data" they are likely referring to "recover 99% of the presence records of each species", right? If that's the case, please, clarify it, as "our data" sounds a bit unspecific here.

*Answer: we agree with the Reviewer's comment and changed the text as they suggest.

Line 220: The sentence "..., represented by the unique deviance explained by each of them." sounds quite unspecific as well. I know the authors are talking about predictors/variables/covariates, so maybe it would make sense to use whatever word they prefer rather than "them".

*Answer: following the Reviewer's suggestion we changed the text as follows:

"we calculated variable importance (Figure 3), both as the total deviance (darker points) explained by each variable and its unique component (i.e. the deviance that was not explained by any other variable, lighter points)."

Line 220: how is the "unique deviance explained" computed for each predictor? I found no mention in the methods (but I haven't read the supplementary material yet).

*Answer: we apologise for not having detailed the variable importance in the methods. We now describe it as follows:

"The relative importance of each environmental variable was quantified for all the models above the BCI threshold of 0.8 in two different ways. Firstly, we computed the total deviance explained by each variable by simply fitting a GAM with only that variable. We then estimated the unique deviance explained by a variable by comparing the full model and a model for which that variable was excluded (i.e. we computed the explained deviance that was lost by dropping that predictor). The difference between the two values represents the deviance explained by a variable which can also be accounted for by other variables (i.e. the deviance in common with other variables)."

Line 308: replace "dispersals" with "dispersal".

*Answer: addressed as suggested

Line 322: replace "strength" with "strengths".

*Answer: the whole paragraph has been removed

Line 322: I tend to disagree with the statement "By using the archaeozoological record, it overcomes the bias given by contemporary ...". Human pressure was already there since at least 300ky, and was likely shaping the distribution and population dynamics of megafauna at that time frame. It is totally true though that the magnitude of the human impact has increased exponentially, but to claim that presence records between 40 and 7.5ky are not affected by human pressure might not be totally safe, more so when the potential impact of human presence, that could be inferred from the archaeological record, has not been assessed here. My point really is "does using the archaeozoological record really help sdms

overcome the bias given by contemporary human impact?", or in other words, if the sentence is assumed as truth, does the palaeo-sdm field benefit from such an assumption, or are we just disregarding a potential source of bias we actually do have data to address?

*Answer: the whole paragraph has been removed

Line 327: When the authors say "Our method, by taking into account all occurrences through time, is less sensitive to this problem...", why is the method less sensitive to undersampling? I cannot see why a lack of presence records during a time-slice would not produce in the GAM an apparent reduction in niche breadth. I agree that the spatio-temporal SDMs fitted in this paper are quite interesting, but to affirm that they can overcome a limitation others method cannot without a proper explanation isn't optimal.

*Answer: the whole paragraph has been removed

Line 330: here the authors affirm that "our approach provides the best possible approximation of the fundamental niche of a species", how so? For example, using true absences (and I understand how difficult is it to extract absences from archaeological contexts) would be a better approach to define niche boundaries than pseudo-absences. My point is that every method has its limitations, and turning a gam into a spatio-temporal sdm doesn't necessarily overcome the limitations sdms already have.

*Answer: the whole paragraph has been removed

Line 392: why are the GAM models evaluated only with the Boyce Continuous Index, but not with their own measures of goodness of fit. The adjusted R squared and the explained deviance should be provided as well. Also, how are the model residuals? Are they normal and spatially autocorrelated? I think all these details would provide the reader with a deeper insight into how good the fitted models are, and in my opinion, the Boyce Continuous Index is not enough in this case.

*Answer: as stated before, we have modified the methods to address this comment. We now perform model choice with AIC, which is the standard test to do it, and also considered explained deviance and Nagelkerke R^2 (that are always in accord with AIC) (Supplementary tables 1-4). Moreover, we now included in the analyses also measures of the spatial autocorrelation in the model residuals (Moran's I, supplementary tables 5-8) and performed standard tests on the residuals (discussed in the text but not shown; the associated code is available in the supplementary material).

Figure 5: How is the variable importance computed? I have seen it in the code, but a few words about it either in the methods section or the figure caption would go a long way.

*Answer: as stated before, we apologise for not having detailed the variable importance in the methods. We now describe it as follows:

"The relative importance of each environmental variable was quantified for all the models above the BCI threshold of 0.8 in two different ways. Firstly, we computed the total deviance explained by each variable by simply fitting a GAM with only that variable. We then estimated the unique deviance explained by a variable by comparing the full model and a model for which that variable was excluded (i.e. we computed the explained deviance that was lost by dropping that predictor). The difference between the two values represents the deviance explained by a variable which can also be accounted for by other variables (i.e. the deviance in common with other variables)."

Thanks for such a great paper, I am looking forward to seeing it published.

Blas M. Benito

Reviewer #2 (Remarks to the Author):

Leonardi et al. use a multi-temporal calibration approach to ecological niche modelling for four European ungulates from the pre-LGM (40kya) to the Holocene (8kya) in order to assess potential niche shifts across this time period. They find that red deer and wild boar did not change niches, as both species have relatively broad climatic niches, while horse and aurochs both expanded their more restricted niches after the LGM and the authors discuss possible reasons, including loss of megafaunal grazing competitors.

In general this is a nice paper, tightly focused on a complementary set of ungulates and making good use of the long-term fossil record to highlight species' long-term niches using a range of statistical approaches.

**Answer: we thank the reviewer for their kind words*

However, there are a few areas where more detail or clarification is required, in particular the methods which I have outlined below in more detail, as it's difficult to assess your results/discussion without fully understanding the process you've gone through with the data.

General comments

**Answer: we have heavily modified the text, so some of the portions to which the comments refer may not be present anymore. For your references, the reviewer's comments have been added next to the associated (even if deleted) text in the manuscript file with marked revisions.*

There are a few places where the semantics around niches gets confusing. In this paper you are really only discussing a species' (bio)climatic niche, or perhaps to be straightforward, its climate envelope. A huge number of different biotic and abiotic factors go towards defining realised niches, not just climate. It may help to clarify by using the term 'climatic envelope' or 'climatic niche' throughout.

**Answer: we thank the Reviewer for pointing out this important aspect. As we added to the analyses a measure of topography, (rugosity, see methods), when needed we used the words "environmental niche"*

Line 46: should this crucially read 'might not be'?

**Answer: both reviewers commented on this and similar statements. After careful consideration we decided to remove them in order to provide a clearer manuscript, more focussed on the analyses and the results than on the methods.*

Line 57-59: a little confusing, do you mean a portion of their fundamental niche here? If human activities are restricting species distributions then is that not a constraint on the fundamental niche and the resultant distribution is the realised niche at that point in time? It may help to change 'realised niche in terms of climatic variables' to 'climatic niche'?

**Answer: the whole paragraph has been removed*

Line 61-64: Again, perhaps I am misunderstanding but by using palaeorecords are you not

helping to more widely define the climate envelope (part of its fundamental niche) of a species rather than its realised niche? Then by constraining the projections using the available climate at each different time point you are plotting the potential realised (climatic) niche at each different time point. If the projections are at a mismatch with fossil records, or other ground-truthing data, then you may then be able to surmise that factors other than bioclimatic variables are constraining the realised niche (as per your discussion)?

*Answer: the whole paragraph has been removed.

Line 102-3: 'niche variation through time' – again, you must be referring to realised niche here.

*Answer: the whole paragraph has been removed.

Line 118-121: can you provide evidence/references as to what the different species' fidelity to 'open' vs 'closed' habitat is based on (dentition, fossil records)? These assumptions underpin most of your interpretation of the data which is interesting given your approach is about not assuming static niches of species. Your results seem to suggest all four species inhabited a variety of habitats as their distributions are very wide!

*Answer: we modified as follows:

“Horse and aurochs are adapted to more open environments (e.g. steppe, grassland, forest-steppe), while deer and wild boar tend to occupy more forested areas (with wild boars mostly favouring broadleaf forests). Moreover, horse and deer are better adapted to the cold within their respective habitat preferences: deer have been more tolerant to different habitats 20–22. It has to be noted that this general picture is also confirmed in prehistoric times when the presence of large mammals in archaeological sites is compared to other proxies 23,24.”

I found Figure 3 and the associated analysis a little confusing to interpret and not sure I followed your results in lines 176-7!

*Answer: to address this comment we added a new figure (Figure 5) and a whole section of the text to better clarify how to interpret figure 6 (formerly “Figure 3”) and the associated analyses:

Figure 1: Schematic description of how the GAM capture the change through time in the effect of an environmental variable. The example is based on the results for minimum temperature in deer.

The geographic distribution of a species over time is affected by the availability of different environmental conditions and the ability of species to use such conditions (i.e. its niche). The nature of the niche change over time is best visualised by plotting the smooths from the GAMs. The smooths represent the effect of a variable on the probability of occurrence. In a changing niche model, the effect changes through time and can be visualised as a heatmap with time on the x-axis and the values of the variable of interest on the y-axis (Figure 1). In such a plot, the heatmap colour shows the effect on the probability of occurrence: red means an increased and blue a decreased probability, and black lines are isoclines. However, note that the final distribution of the species will

be the product of these probabilities and the relative abundance of different environmental conditions. For example, if we had a variable with no effect (corresponding to a completely white smooth plot), we would still expect to see the species to be most commonly found in the most represented values for the environmental variable of interest. Availability is critical in determining the final distribution: a prevalent but less preferred environmental condition might still harbour a large portion of the range of a species compared to a favoured but scarce condition. Thus, to understand the impact of the changes in habitat suitability given by the smooths, we need to inspect them in conjunction with the occurrences and the changes in available environmental space. Finally, note that the variable with the largest change might not be the most important one in determining the overall distribution (see the earlier section on variable importance).

Line 199: it looks from Figure 4 like most species expanded or maintained their ranges into the LGM? Do you have a quantification of range that supports your assertion that ranges contracted?

*Answer: the whole paragraph has been removed.

Lines 199-204: northern limits of ranges seem to only constrained by the ice sheets in most cases?

*Answer: we thank the reviewer for this insightful comment, we added this statement to the main text

Line 214: I cannot find Figure 5A so cannot assess this part of the results.

*Answer: We thank the Reviewer for having spot this mistake. This portion of text refers to a figure that has been deleted from the manuscript at an earlier stage. We have now deleted the whole paragraph, and we apologise for not having noticed this error before.

Lines 232-233: Can you explain this? I suspect it's because productivity of vegetation (herbs) is important for grazers, whereas precipitation influences shrub/forest growth?

*Answer: the whole paragraph has been removed.

Line 240-242: But from your maps it looks like the aurochs does inhabit the Caucasus pre-LGM and LGM, and the horse in LGM? There is a relatively good Holocene fossil record for these species in the Caucasus too.

*Answer: we modified the sentence into "On the other hand, the higher-altitude areas in the Caucasus show an opposite pattern, as they become more unsuitable for aurochs and wild boar starting from the LGM and for the horse in the Holocene".

Line 265-6: Could this be due to confusion between domestic and wild horse i.e. potentially a lot of wild horse not identified or down as Equus sp?

*Answer: currently, the archaeological/scientific community agrees about a marked reduction in horses remain in this period (which is prior to the attested beginning of their domestication). We added a few references to support this statement:

"horses (not adapted to forested areas) were relatively rare before domestication when the others were abundant 11,27–30."

Line 289: 'vegetal' = plant?

*Answer: the whole paragraph has been removed.

Methods

Much more detail is needed in the methods around primary data collection, as what is fed into the models will be highly influential to model performance and predictions. It would also be very hard to replicate this study without more information on data collection.

*Answer: we acknowledge that the whole section is confusing, so we reformulated it entirely. At the same time, we would like to highlight that in order to ensure full reproducibility of the data the full datasets and codes used for the analyses have been (and will remain, upon publication) shared through figshare (link: <https://figshare.com/s/ffb398b52ec27d6bb425>)

The section now reads:

“We collected from the literature and available databases a dataset of radiocarbon dates from Europe (West of 60°E and North of 37°N) either obtained from remains of the four analysed species, or from archaeological layers where the species has been observed. We only considered observations dated between 7,500 and 47,000 cal BP: their scarcity before this period may bias the GAMs, and after it, domesticated cattle, pigs and (later) horses arrived in Europe, making it difficult to differentiate them from their wild forms.

We excluded any record fitting one or more of the following conditions: unreliable; not in accord with the expected chronology of their archaeological layer; without a reported standard error; available only as terminum ante/post quem.

All dates were calibrated with OxCal 5 version 4.4 using the IntCal20 curve 48, and we further excluded any record for which calibration resulted in an error, resulting in the number of points presented in Table 1 as "Original dataset" (available in the supplementary material).

SDMs based on GAMs need presence/background data, not frequencies; moreover, multiple observations (i.e. presence in different archaeological layers) from the same site and time slice are likely to introduce stronger sample biases linked to chrono-geographically differential sampling efforts. For this reason, we collapsed our observations by keeping only one point per grid cell per time slice for each species, leaving the number of observations reported in Table 1 as "Collapsed datasets", used for all the analyses presented in this work (and available in the supplementary material).”

Line 343-344: ‘whilst for the other three species we collected direct and indirect radiocarbon dates from the literature’ – far more information is needed on how radiocarbon dates were collected, as online databases notoriously have a large amount of unreliable data that has been uploaded by multiple people and not checked. For example, which online databases were used? What were the criteria for selecting ‘good’ i.e. reliable dates? What were the criteria for selecting indirect dates? There is huge potential in indirect dates for including data that do not relate to your focal species without clearly thought out auditing criteria.

*Answer: please refer to the previous comment

Line 347: ‘all dates were...then classified into time slices’. Do these time slices refer to the final four time periods used to present the data in the paper, or the 1000-2000 year time slices provided by the bioclimatic data that are presented in the SI?

*Answer: We have clarified this issue by rewriting the sentence:

“To perform all analyses, we coupled each observation from the collapsed datasets to paleoclimatic reconstructions published in 8, which are based on the Hadley CM3 model,

include 14 different bioclimatic variables at a spatial resolution of 0.5° and are available for the whole world every 1,000 years until 22 kya and every 2,000 years before that date (referred in the manuscript as "time slices"). Specifically, each observation was associated with the relevant bioclimatic reconstruction based on its average age and spatial coordinates."

How were data classified into the time slices e.g. did you use the mid-point of each calibrated radiocarbon date; the weighted mean; did you include a record in every time slice that it fell?

*Answer: as for the previous comment, we changed the sentence to include this information:

"To perform all analyses, we coupled each observation from the collapsed datasets to paleoclimatic reconstructions published in 8, which are based on the Hadley CM3 model, include 14 different bioclimatic variables at a spatial resolution of 0.5° and are available for the whole world every 1,000 years until 22 kya and every 2,000 years before that date (referred in the manuscript as "time slices"). Specifically, each observation was associated with the relevant bioclimatic reconstruction based on its average age and spatial coordinates."

Line 351: Describing the 'cleaning' of this dataset here is very confusing as it turns out to be the 'raw' dataset, and not the one you go on to call 'cleaned' (line 360)! Could the language here be adjusted for clarification?

*Answer: please refer to the answer to the general comment of the "methods" section

Line 351-353: Was this the only cleaning of data that occurred? Did you double check species IDs, that dates were from a reliable lab, that a date wasn't old and/or hadn't been superseded by re-dating, did you check references etc? There is a large amount of 'dodgy' dates in online databases so it's really important to demonstrate at this stage that you've taken steps to ensure your data is as reliable as possible.

*Answer: please refer to the answer to the general comment of the "methods" section

Line 359: Why did you collapse the data (I assume to reduce spatial bias but it would be helpful to say why)? Did you do this for each species?

*Answer: we added this explanation to the text as follows:

"SDMs based on GAMs need presence/background data, not frequencies; moreover, multiple observations (i.e. presence in different archaeological layers) from the same site and time slice are likely to introduce stronger sample biases linked to chrono-geographically differential sampling efforts. For this reason, we collapsed our observations by keeping only one point per grid cell per time slice for each species, leaving the number of observations reported in Table 1 as "Collapsed datasets", used for all the analyses presented in this work (and available in the supplementary material)."

Lines 361-367: Why were these variables chosen i.e. what was the biological reasoning? And again, how were they linked temporally to calibrated radiocarbon dates?

*Answer: to answer this comment, we modified the text as follows:

"As already mentioned, the four species analysed show different preferences in terms of temperature, habitat and altitude. For the Species Distribution Modelling, we then choose five environmental variables that should be able to capture such differences: two measures of temperature (BIO5, maximum temperature of the warmest month, and BIO6, minimum temperature of the coldest month); two variables to help capture habitat differentiation

(BIO12 annual precipitation, and Net Primary Productivity, NPP), and one measure of topography (rugosity).”

Line 385-6: Can you clarify how ‘time’ was used as a variable as I am not familiar with tensor products and cannot visualise how it was used in the model?

*Answer: to answer this comment, we added the following explanation in the methods:

“In a univariate GAM, the effect of a variable is modelled as a smooth, which allows its effect to take a complex form (a polynomial is a simple way to think about it). So, if we consider the effect of temperature at a given time, we can imagine a situation where a species is more likely to be found at intermediate temperatures. When considering the change of preference for a given temperature over time, it can also be visualised as a smooth (at intermediate temperature, we could find a preference early on that becomes avoidance later). A tensor product brings these two together into a single function (technically, the score for a given combination of temperature and time is a product of functions). Thus, the tensor product between temperature and time allows that relationship to progressively change over time (so, the preference for intermediate temperature could eventually become a preference for high temperature, or even a preference for both high and low temperatures with an avoidance of intermediate ones). The mechanics of the tensor product allows us to estimate the appropriate amount of “wiggleness” of the functions that best explains the data without overfitting (i.e. without selection functions that completely match every datapoint in the temperature by time space).”

Reviewers' comments:

Reviewer #3 (Remarks to the Author):

Dear Editor and Authors,

I finished reviewing the paper "Following the niche: the differential impact of the Last Glacial Maximum on four European ungulates". As a foreword, I don't know much about the four species and their past distribution, so I approached this review as a modeler rather than as an animal ecologist.

The study focuses on reconstructing past environmental niches for four species of ungulates in Europe employing a general additive model (GAM) approach. This is a quite extensive and complex method (which I still have to fully master) and the authors did a great job in conveying some of the most complicated aspect of their methodology in a concise, yet informative way. The main aim and contribution of the study is to show if environmental niche of the four species has changed over time during the last 50,000 years and how this can be accounted for in species distribution modeling (SDM). This is a quite ambitious goal, motivated by the current knowledge gap of: how frequent niche changes are? how to model them? The methods used in this study fit the problem, the manuscript is well-written and clear, the results are interpreted in an objective way and the authors' arguments are convincing.

I have gone through the comments from the previous round of review and I believe that the authors did accommodate all reviewers requests. Reviewer #2 had some suggestions on the semantic of the term "niche" used here; I'm not surprised as there are as many definitions of niches as ecologists, but I tend to agree that the semantic can be slightly improved to make the interpretation of the GAM easier.

I think that what the GAM with the time term does is to reconstruct a fundamental environmental niche that is less-biased compared to a GAM without time. Such biases emerge because of uneven sampling (both in number and predictor range) across time slices. This uneven sampling can be due to the species not being present or because of sampling biases, but in the case of this study I don't think it's important the cause (this is very well explained at lines 89-96 - great paragraph to summarize the core motivation of using GAMs - and in figure 1). I think that the closest term used by the authors to call this less-biased niche is 'potential distribution'. When also the temporal term is plotted, however, I am not sure this is still a potential niche, as the temporal term, or better its interaction with the environmental covariates, will modify the time-specific GAM (SDM) to fit the data of the specific time slice. In other words, I believe that the potential distribution can be plotted when considering only the marginal response due to environmental covariates, while if time is included, this will already constrain species' distribution to a 'realized' niche. This does not matter much in term of results, but it does a bit for the terminology. For instance, at line 193 and 198 the term 'potential distribution' is confusing to me, as these seem to me to be already realized distributions, i.e. where other abiotic and biotic factors constrained the potential distribution (as estimated by GAMs). I am not entirely sure I am right, and any clarification is welcome.

I found a bit odd some steps of the GAM. I'm not too concerned about the validity of results, but there are few things that I would change for correctness:

1. Detection of high collinearity should account also for non-linear correlation. Pearson's correlation does not account for this, and `mgcv::concurvity` should be used instead. The drawback is that there' is not a clear-cut threshold for it and it's more subjective to interpretations. I don't think this is a big deal, but because the whole point of GAM is to use non-linear smoothers, it seems odd to than test if they were associated by linear correlation only.
2. The main practical reason to use `te()` instead `s()` is because the covariates have different scales (see *Generative Additive Models – An Introduction in R*; Wood, 2006; p. 228). I suggest this to be clarified and that some unnecessary sentences at lines 420-435 to be removed.

From:

"In a univariate GAM, the effect of a variable is modelled as a smooth, which allows its effect to take a complex form (a polynomial is a simple way to think about it). So, if we consider the effect of temperature at a given time, we can imagine a situation where a species is more likely to be found at intermediate temperatures. When considering the change of preference for a given temperature over time, it can also be visualised as a smooth (at intermediate temperature, we could find a preference early on that becomes avoidance later). A tensor product brings these two together into a single function (technically, the score for a given combination of temperature and time is a product of functions). Thus, the tensor product between temperature and time allows that relationship to progressively change over time (so, the preference for intermediate temperature could eventually become a preference for high temperature, or even a preference for both high and low temperatures with an avoidance of intermediate ones). The mechanics of the tensor product allows us to estimate the appropriate amount of "wiggleness" of the functions that best explains the data without overfitting (i.e. without selection functions that completely match every datapoint in the temperature by time space)."

To:

"In a univariate GAM, the effect of a variable is modelled as a smooth, which allows its effect to take a complex form. When considering the change of preference for a given temperature over time, it can also be visualised as a smooth (a complex ...). A tensor product brings these two together into a single function by allowing covariates to have different scales. Thus, the tensor product between temperature and time allows that relationship to progressively change over time."

3. From the previous paragraph and subsequent lines: "The mechanics of the tensor product allows us to estimate the appropriate amount of "wiggleness" of the functions that best explains the data without overfitting (i.e. without selection functions that completely match every datapoint in the temperature by time space). For all GAMs, we set 4 as the maximum threshold for the degrees of freedom of the splines; this value provides a reasonable compromise between allowing the relationship to change through time but avoiding excessive overfitting". This is not a property of `te()` per se and I found odd to choose $k = 4$ as basis dimension to avoid overfitting. There are formal ways to test the optimal k , e.g. looking at GCV score or `gam.check()`, and extra-penalty to the smoothers can be added with `gam(gamma = 1.4)`, as recommended (Woods, p. 231). I don't think an a-prior $k = 4$ is well-justified in general, as optimal k is highly problem-specific. I don't have the data to run the analysis, but I guess in any case that $k = 4$ should be on the reasonable end for all covariates.

4. I would explicitly state that thin plate regression splines (TPNS; `bs = "tp"`) was used (even if `mgcv` default).

5. I did not understand the lines 214-226. I didn't encounter this argument before and there are not enough details or reference to guide me on this.

Of all points above, the most critical to me is point 5, as this can influence the maps more than the other points and it is not sufficiently explained or references provided. I don't mind too much the other points; even if they may promote some practices that are not entirely correct, GAM should be sufficiently robust to provide similar, if not identical, results.

As a final word, I would like to state again that I enjoyed very much reading the paper and was very impressed by the ability of the authors to communicate some of the most complex topics in the study in a simple and communicative way. It is a very interesting, well-thought study that is communicated very efficiently and that made me think quite a lot. Overall, the authors illustrated a way to model species' distribution through time that is reliable, less-biased, and relatively easy to perform compared to other approaches. This is a great contribution and one that I am looking forward to see published.

I finished reviewing the paper “Following the niche: the differential impact of the Last Glacial Maximum on four European ungulates”. As a foreword, I don’t know much about the four species and their past distribution, so I approached this review as a modeler rather than as an animal ecologist.

The study focuses on reconstructing past environmental niches for four species of ungulates in Europe employing a general additive model (GAM) approach. This is a quite extensive and complex method (which I still have to fully master) and the authors did a great job in conveying some of the most complicated aspect of their methodology in a concise, yet informative way. The main aim and contribution of the study is to show if environmental niche of the four species has changed over time during the last 50,000 years and how this can be accounted for in species distribution modeling (SDM). This is a quite ambitious goal, motivated by the current knowledge gap of: how frequent niche changes are? how to model them? The methods used in this study fit the problem, the manuscript is well-written and clear, the results are interpreted in an objective way and the authors’ arguments are convincing.

***Answer: we thank the reviewer for their kind words**

I have gone through the comments from the previous round of review and I believe that the authors did accommodate all reviewers requests. Reviewer #2 had some suggestions on the semantic of the term “niche” used here; I’m not surprised as there are as many definitions of niches as ecologists, but I tend to agree that the semantic can be slightly improved to make the interpretation of the GAM easier.

I think that what the GAM with the time term does is to reconstruct a fundamental environmental niche that is less-biased compared to a GAM without time. Such biases emerge because of uneven sampling (both in number and predictor range) across time slices. This uneven sampling can be due to the species not being present or because of sampling biases, but in the case of this study I don’t think it’s important the cause (this is very well explained at lines 89-96 - great paragraph to summarize the core motivation of using GAMs - and in figure 1). I think that the closest term used by the authors to call this less-biased niche is ‘potential distribution’. When also the temporal term is plotted, however, I am not sure this is still a potential niche, as the temporal term, or better its interaction with the environmental covariates, will modify the time-specific GAM (SDM) to fit the data of the specific time slice. In other words, I believe that the potential distribution can be plotted when considering only the marginal response due to environmental covariates, while if time is included, this will already constrain species’ distribution to a ‘realized’ niche. This does not matter much in term of results, but it does a bit for the terminology. For instance, at line 193 and 198 the term ‘potential distribution’ is confusing to me, as these seem to me to be already realized distributions, i.e. where other abiotic and biotic factors constrained the potential distribution (as estimated by GAMs). I am not entirely sure I am right, and any clarification is welcome.

***Answer: we have modified the text to remove the confusing terms**

I found a bit odd some steps of the GAM. I’m not too concerned about the validity of results, but there are few things that I would change for correctness:

1. Detection of high collinearity should account also for non-linear correlation. Pearson’s correlation does not account for this, and `mgcv::concurvity` should be used instead. The drawback is that there’ is not a clear-cut threshold for it and it’s more subjective to

interpretations. I don't think this is a big deal, but because the whole point of GAM is to use non-linear smoothers, it seems odd to than test if they were associated by linear correlation only.

***Answer:** we have taken into account this suggestion calculating non-linear correlation with `mgcv::concurvity`, as detailed in the following paragraph in the methods

“We checked for non-linear correlation among variables using the `mgcv::collinearity` function and checked the values of estimated concurvity. All estimates were below the threshold of 0.8 in all models, run and variables except for a few instances for time. We consider this not to be worrying: this is most likely a result of sample bias, and GAM is known to be robust to correlation/concurvity^{55,56}.”

2. The main practical reason to use `te()` instead `s()` is because the covariates have different scales (see Generative Additive Models – An Introduction in R; Wood, 2006; p. 228). I suggest this to be clarified and that some unnecessary sentences at lines 420-435 to be removed.

From:

“In a univariate GAM, the effect of a variable is modelled as a smooth, which allows its effect to take a complex form (a polynomial is a simple way to think about it). So, if we consider the effect of temperature at a given time, we can imagine a situation where a species is more likely to be found at intermediate temperatures. When considering the change of preference for a given temperature over time, it can also be visualised as a smooth (at intermediate temperature, we could find a preference early on that becomes avoidance later). A tensor product brings these two together into a single function(technically, the score for a given combination of temperature and time is a product of functions). Thus, the tensor product between temperature and time allows that relationship to progressively change over time (so, the preference for intermediate temperature could eventually become a preference for high temperature, or even a preference for both high and low temperatures with an avoidance of intermediate ones). The mechanics of the tensor product allows us to estimate the appropriate amount of “wiggleness” of the functions that best explains the data without overfitting (i.e. without selection functions that completely match every datapoint in the temperature by time space).”

To:

“In a univariate GAM, the effect of a variable is modelled as a smooth, which allows its effect to take a complex form. When considering the change of preference for a given temperature over time, it can also be visualised as a smooth (a complex ...). A tensor product brings these two together into a single function by allowing covariates to have different scales. Thus, the tensor product between temperature and time allows that relationship to progressively change over time.”

***Answer:** After careful consideration of the reviewer comment and suggestion, we decided to change the text as follows:

“In GAMs, the effect of a given continuous predictor on the response variable (in our case, the logit transformed probability of a presence) is represented by a smooth function; this smooth function can be linear or non-linear and can become highly complex in shape depending on the number of knots selected by the GAM fitting

algorithm. The interaction between two covariates is modelled by tensor products; this approach is equivalent to an interaction term in a linear model, but with the added complexity of the smooth function. In our models, we confine tensor products to the interaction between an environmental variable and time; a simple way to think about such a tensor product is that it allows the smooth representing the relationship between the variable and the probability of a presence to change progressively over time.”

3. From the previous paragraph and subsequent lines: “The mechanics of the tensor product allows us to estimate the appropriate amount of “wiggleness” of the functions that best explains the data without overfitting (i.e. without selection functions that completely match every datapoint in the temperature by time space). For all GAMS, we set 4 as the maximum threshold for the degrees of freedom of the splines; this value provides a reasonable compromise between allowing the relationship to change through time but avoiding excessive overfitting”. This is not a property of `te()` per se and I found odd to choose $k = 4$ as basis dimension to avoid overfitting. There are formal ways to test the optimal k , e.g. looking at GCV score or `gam.check()`, and extra-penalty to the smoothers can be added with `gam(gamma = 1.4)`, as recommended (Woods, p. 231). I don’t think an a-prior $k = 4$ is well-justified in general, as optimal k is highly problem-specific. I don’t have the data to run the analysis, but I guess in any case that $k = 4$ should be on the reasonable end for all covariates.

***Answer:** we have modified the analyses as suggested. Here is the new description from the method’s section.

“The GAM algorithm automatically selects the complexity of the smooth most appropriate to the data that are being fitted; as GAM can have issues with overfitting, we added an additional penalty against overly complex smooths ($\text{gamma}=1.4$) and used Restricted Maximum Likelihood (REML=TRUE), as recommended by⁵⁴. It is possible that even with these settings, the complexity of the smooth is not sufficient; we used `mgcv::gam.check()` to check this, and increased the basis dimension of the smooth, k , to make sure that $k-1$ was larger than the estimated degrees of freedom (edf). We found the best maximum thresholds for k to be 16 for bio06 and 10 for all other variables.”

4. I would explicitly state that thin plate regression splines (TPNR; `bs = “tp”`) was used (even if `mgcv` default).

***Answer:** we added this information to the main text as follows:

GAMs were fitted using the `mgcv` package in R⁵⁴ using thin plate regression splines (TPNR; `bs = “tp”`, default in `mgcv`) for environmental variables, and their tensor products with time in the “niche changing” models.

5. I did not understand the lines 214-226. I didn’t encounter this argument before and there are not enough details or reference to guide me on this.

***Answer:** we thank the reviewer for spotting a portion of the text that wasn’t clear enough to the reader. We have heavily modified it and added a reference for the interaction plot to make it more understandable.

“The easiest way to visualise the change in the niche through time is to use the interaction plots as generated by the R package gratia27 (figure 5).

The idea behind it is that the geographic distribution of a species over time (i.e., prediction) is affected by the availability of different environmental conditions (i.e., climate) and the ability of species to use such conditions (i.e., its niche). The prediction of the species’ distribution should be considered as the product of the relative abundance of different environmental conditions available in the area (“climate”) and the GAM (“the effect sizes shown in the interaction plot”), detailing the nature of the niche change over time where the smooths represent the effect of a variable on the predicted probability of occurrence. Because in a changing niche model this effect changes through time, it can be visualised as a heatmap with time on the x-axis and the values of the variable of interest on the y-axis. In such a plot, the heatmap colour shows the effect on the probability of occurrence: red means an increased and blue a decreased probability compared to what would be expected based on the distribution of values for a given variable; and black lines are isoclines.

If we had a variable with no effect (corresponding to a completely white smooth plot), we would still expect to see the species to be found most commonly in the most represented values for the environmental variable of interest. Availability is critical in determining the final distribution: a prevalent but less preferred environmental condition might still harbour a large portion of the range of a species compared to a favoured but scarce condition. Thus, to understand the impact of the changes in habitat suitability given by the smooths, we need to inspect them in conjunction with the occurrences and the changes in available environmental space. Finally, note that the variable with the largest change might not be the most important one in determining the overall distribution (see the earlier section on variable importance).”

Of all points above, the most critical to me is point 5, as this can influence the maps more than the other points and it is not sufficiently explained or references provided. I don’t mind too much the other points; even if they may promote some practices that are not entirely correct, GAM should be sufficiently robust to provide similar, if not identical, results.

As a final word, I would like to state again that I enjoyed very much reading the paper and was very impressed by the ability of the authors to communicate some of the most complex topics in the study in a simple and communicative way. It is a very interesting, well-thought study that is communicated very efficiently and that made me think quite a lot. Overall, the authors illustrated a way to model species’ distribution through time that is reliable, less-biased, and relatively easy to perform compared to other approaches. This is a great contribution and one that I am looking forward to see published.

Sincerely,
Emilio Berti